



**History of anthropogenic Nitrogen inputs (HaNi) to the terrestrial biosphere:**
**A 5-arcmin resolution annual dataset from 1860 to 2019**
Hanqin Tian[1]\*, Zihao Bian[1]\*, Hao Shi[2,1]\*, Xiaoyu Qin[2], Naiqing Pan[1], Chaoqun Lu[3], Shufen
Pan[1], Francesco N. Tubiello[4], Jinfeng Chang[5], Giulia Conchedda[4], Junguo Liu[6], Nathaniel
Mueller[7,8], Kazuya Nishina[9], Rongting Xu[10], Jia Yang[11], Liangzhi You[12], Bowen Zhang[13]
[1]International Center for Climate and Global Change Research and School of Forestry and
Wildlife Sciences, Auburn University, Auburn, AL 36849, USA; [2]Research Center for Eco-
Environmental Sciences, State Key Laboratory of Urban and Regional Ecology, Chinese
Academy of Sciences, Beijing 100085, China; [3]Department of Ecology, Evolution, and
Organismal Biology, Iowa State University, Ames, IA 50011, USA; [4]Statistics Division, Food
and Agriculture Organization of the United Nations, Via Terme di Caracalla, Rome, Italy;
[5]College of Environmental and Resource Sciences, Zhejiang University, Hangzhou 310058,
China; [6]School of Environmental Science and Engineering, Southern University of Science and
Technology, Shenzhen 518055, China.[7]Department of Ecosystem Science and Sustainability,
Colorado State University, Fort Collins, CO 80523, USA;[8]Department of Soil and Crop
Sciences, Colorado State University, Fort Collins, CO 80523, USA; [9]Biogeochemical Cycle
Modeling and Analysis Section, Earth System Division, National Institute for Environmental
Studies 16-2, Onogawa, Tsukuba, 305-8506, JAPAN; [10]Forest Ecosystems and Society, Oregon
State University, Corvallis, OR 97330, USA; [11]Department of Natural Resource Ecology and
Management, Oklahoma State University, Stillwater, OK 74078, USA;[12]International Food
Policy Research Institute (IFPRI), 1201 Eye Street, NW, Washington, DC 20005, USA;
[13]Department of Environment, Geology, and Natural Resources, Ball State University, Muncie,
IN 47306, USA

*Corresponding authors*:

Hanqin Tian (tianhan@auburn.edu);

Zihao Bian (zzb0009@auburn.edu);

Hao Shi (haoshi@rcees.ac.cn)





**Abstract**
Excessive anthropogenic nitrogen (N) inputs to the biosphere have disrupted the global nitrogen
cycle. To better quantify the spatial and temporal patterns of anthropogenic N enrichments, assess
their impacts on the biogeochemical cycles of the planet and other living organisms, and improve
nitrogen use efficiency (NUE) for sustainable development, we have developed a comprehensive
and synthetic dataset for reconstructing the History of anthropogenic N inputs (HaNi) to the
terrestrial biosphere. The HaNi dataset takes advantage of different data sources in a
spatiotemporally consistent way to generate a set of high-resolution gridded N input products from
the preindustrial to present (1860-2019). The HaNi dataset includes annual rates of synthetic N
fertilizer, manure application/deposition, and atmospheric N deposition in cropland, pasture, and
rangeland at a spatial resolution of 5-arcmin. Specifically, the N inputs are categorized, according
to the N forms and land uses, as ten types: 1) $NH_4^+$-N fertilizer applied to cropland, 2) $NO_3$-N
fertilizer applied to cropland, 3) $NH_4^+$-N fertilizer applied to pasture, 4) $NO_3$-N fertilizer applied
to pasture, 5) manure N application on cropland, 6) manure N application on pasture, 7) manure
N deposition on pasture, 8) manure N deposition on rangeland, 9) $NH_x$-N deposition, and 10) $NO_y$-
N deposition. The total anthropogenic N (TN) inputs to global terrestrial ecosystems increased
from 29.05 Tg N yr$^{-1}$ in the 1860s to 267.23 Tg N yr$^{-1}$ in the 2010s, with the dominant N source
changing from atmospheric N deposition (before the 1900s) to manure N (the 1910s-2000s), and
to synthetic fertilizer in the 2010s. The proportion of synthetic $NH_4^+$-N fertilizer increased from
64% in the 1960s to 90% in the 2010s, while synthetic $NO_3$-N fertilizer decreased from 36% in
the 1960s to 10% in the 2010s. Hotspots of TN inputs shifted from Europe and North America to
East and South Asia during the 1960s-2010s. Such spatial and temporal dynamics captured by the
HaNi dataset are expected to facilitate a comprehensive assessment of the coupled human-earth
system and address a variety of social welfare issues, such as climate-biosphere feedback, air
pollution, water quality, and biodiversity.


## 1. Introduction

Nitrogen (N) is an essential element for the survival of all living organisms, required by various biological molecules, for instance, nucleic acids, proteins, and chlorophyll (Galloway et al., 2021; Schlesinger and Bernhardt, 2020). Most N on the Earth is not readily available for organisms, since it either exists in the form of inert $N_2$ gas or is stored in crust and sediments (Ward, 2012). Driven by the human demand for food and energy, a spectrum of approaches have been developed to produce biologically available N (Sutton et al., 2013; Lassaletta et al. 2016), ranging from traditional methods, such as legume crops cultivation and manure application, to modern techniques, such as industrial compost and the Haber-Bosch process that produce organic fertilizer mixture and chemical fertilizer. Increasing anthropogenic N inputs have significantly boosted crop yield and improved food security (Stewart and Roberts, 2012), but also resulted in over twofold increase in terrestrial reactive N (Galloway and Cowling 2002; Fowler et al. 2013; Melillo, 2021; Scheer et al., 2020) and are expected to continually increase in the coming decades due to human demand for food (Kanter et al. 2020; Sutton et al. 2021).

The large amount of excessive reactive N in terrestrial ecosystems has led to multiple environmental issues like water quality deterioration, air pollution, global warming, and biodiversity loss (Bouwman et al., 2005; Gruber and Galloway, 2008; Howarth, 2008; Pan et al., 2021; Tian et al., 2020a; Vitousek et al., 1997). The river export of various forms of nitrogen (ammonium, nitrate, dissolved organic N) has largely increased (Schlesinger et al., 2006; Tian et al., 2020b), frequently causing large-scale hypoxia along coastal oceans for example, in the northern Gulf of Mexico (Bargu et al., 2019; Dodds, 2006; Rabalais and Turner, 2019). The global emission of ammonia ($NH_3$), a toxic pollutant and a major precursor of aerosol, had rapidly increased from 1.0 Tg N yr$^{-1}$ in 1961 to 9.9 Tg yr$^{-1}$ in 2010, mainly due to the wide use of N fertilizer (Xu et al., 2019a). The emissions of nitrous oxide ($N_2O$), the third most important greenhouse gas, had increased by 30% over the past four decades, which was mainly attributed to N addition to croplands (Cui et al., 2021; Tian et al., 2020a). Moreover, excessive usage of N over other nutrients (e.g. phosphorus) brings nutrient imbalance that may induce significant alterations in the structure and functions of ecosystems and finally result in losses of biodiversity (Galloway et al., 2003; Lun et al., 2018; Peñuelas and Sardans, 2022; Houlton et al., 2019).



In light of the critical impacts of N excess on the human-earth system, numerous efforts have been
conducted to generate distribution maps of N inputs for different sectors with varied temporal
coverage and spatial resolution (Potter et al., 2010; Nishina et al., 2017; Bian et al., 2021; Liu et
al., 2010). Country-level N fertilizer data from the Food and Agriculture Organization of the
United Nations (FAO) and the International Fertilizer Association (IFA) have been widely used to
assess global and national nitrogen budgets for crop production (Xiong et al., 2008; Zhang et al.,
2021; Eickhout et al., 2006). However, spatial variations of N inputs within countries have been
overlooked in country-level data, while detailed geospatial distributions of N inputs are required
for many process-based modeling studies (Tian et al., 2019, 2018). Potter et al. (2010) and Mueller
et al. (2012) both generated crop-specific spatially-explicit N fertilizer data which, however,
represented the average fertilizer application patterns around 2000. Liu et al. (2010) developed a
N balance model, and made the first attempt to quantify six N inputs (e.g. mineral fertilizer, manure,
atmospheric deposition, biological fixation, input from sedimentation, and input from recycled
crop residual) and five N outputs (e.g. output to harvested crops, crop residues, leaching, gaseous
losses, and soil erosion) in cropland for the year 2000 with a spatial resolution of 5-arcmin. Lu and
Tian (2017) created an annual dataset of global N fertilizer application in cropland at a spatial
resolution of $0.5° \times 0.5°$ during 1961-2013, and Nishina et al. (2017) further split synthetic N
fertilizer application into $NH_4^+$ and $NO_3^-$ forms. Meanwhile, Zhang et al. (2017) reconstructed
global manure N production and application rates in cropland which covered the period 1860-2014
and had a resolution of 5-arcmin; using a similar methodology, Xu et al., (2019b) further developed
three gridded datasets, i.e., rangeland manure deposition, pasture manure deposition, and pasture
manure application, all of which had a resolution of $0.5° \times 0.5°$ and spanned from 1860-2016.
Although these datasets are valuable in addressing their respective objective issues, there is a
barrier in taking advantage of them simultaneously, due to the inconsistent temporal coverage,
spatial resolution, data sources (e.g., N inputs statistics and land use), and spatial allocation
algorithms. Therefore, the reconstruction for the History of Anthropogenic N Inputs (HaNi) to the
terrestrial biosphere with rich spatial details and long-term coverage is essentially needed.
To address this issue, using sophisticated methodologies, we employed multiple statistical data,
empirical estimates, atmospheric chemistry model outputs (Eyring et al., 2013), and high-
resolution land-use products to generate the HaNi dataset. This comprehensive dataset consists of
N fertilizer/manure application to cropland, manure application/deposition to pasture, manure



deposition to rangeland, and atmospheric N deposition on all agricultural land at a resolution of 5-
arcmin from 1860 to 2019. Additionally, we tried to investigate the impacts of social-economic
forcing on N use across different regions. These efforts are anticipated to benefit understanding
the spatial and temporal patterns of human-induced N enrichment, assessing impacts of excessive
N on global and regional biogeochemical cycles, and providing data support for resource
management. The HaNi dataset has also been expected to serve as input data for Earth system
models, biogeochemical models, and hydrological models for improving our understanding and
assessment of global consequences of anthropogenic nitrogen enrichment for climate change, air
and water quality, ecosystems, and biodiversity (e.g. Tian et al., 2018).
**2. Methods**
**2.1. Data sources of fertilizer/manure use**
Multiple anthropogenic N input databases were integrated to generate the HaNi dataset (Table 1).
For the period of 1961-2019, annual country-level statistics data was obtained from the FAOSTAT
"Land, Inputs and Sustainability" domain (FAO, 2021). "N fertilizer applied to soil" was from the
"Fertilizers by Nutrient" subsection. "Manure applied to soil" and "Manure left on pasture" data
were from the "Livestock Manure" subsection. Before 1961, the time series of fertilizer and
manure use from Holland et al. (2005) was adopted and corrected to be consistent with FAO
statistics. For countries (e.g., the former Soviet Union, the Socialist Federal Republic of
Yugoslavia, Eritrea, Ethiopia, and the Czechoslovak Republic) that experienced political
disintegration, we partitioned their pre-disintegration N fertilizer/manure use into each individual
new-formed country using the ratios derived from the N uses of the new-formed countries in the
first year after disintegration.
The FAOSTAT agricultural use of N fertilizer and manure referred to the N use for crops, livestock,
forestry, fisheries, and aquaculture, excluding N use for animal feed. Since the use of N fertilizers
and manure for forestry, fisheries, and aquaculture was minor compared to that for crops and
livestock, this part was taken as neglectable. The partitioning ratio of N fertilizer application to
cropland and pasture was adopted from Lassaletta et al. (2014). Since Lassaletta's ratio values only
covered the period of 1961-2009, values in 2009 were used to calculate the N application
partitioning after 2009. By FAO's definition, manure applied to soil was equal to the difference
between all treated manure and N loss during stored and treated processes. Therefore, we assumed



that the total quantity of manure applied to soil was equal to the total quantity of manure applied
to cropland and pasture. The fraction values for cropland were from Zhang et al. (2015), who
assumed the fraction value ranged between 0.5 and 0.87 for European countries, Canada, and the
U.S., while it was 0.9 for other countries.
**2.2 Land use data**
The HYDE3.2 dataset (Klein Goldewijk et al., 2017) provides historical spatial distributions of
cropland, pasture, and rangeland at a 5-arcmin resolution and at an annual time-step after 2000 but
a decadal time-step before the 1990s. In contrast, the LUHv2 dataset (Hurtt et al., 2020), derived
mainly from HYDE3.2, has an annual time-step across 1860-2019 but at a relatively low spatial
resolution of $0.25° \times 0.25°$. To reconcile these two datasets, we first conducted a linear
interpolation to HYDE3.2 before 1999 using the data of every two neighbor decades. Then the
fraction of crop/pasture/rangeland of a LUHv2 grid was partitioned into all grid cells of HYDE3.2
that fell in the LUHv2 grid, according to their shares in HYDE3.2. Through this routine, we
obtained a land-use dataset that both kept spatial information of HYDE3.2 and was consistent with
LUHv2 on the total area for each land use type.
**2.3 Spatializing N fertilizer and manure application in cropland**
The workflow of spatializing the country-level N fertilizer and manure use amount to gridded maps
is shown in Fig 1. First, the grid-level crop-specific N fertilizer and manure use rates per cropland
area of 17 dominant crop types (wheat, maize, rice, barley, millet, sorghum, soybean, sunflower,
potato, cassava, sugarcane, sugar beet, oil palm, rapeseed, groundnut, cotton, and rye), which were
developed by Mueller et al. (2012) and West et al. (2014), were combined with the crop-specific
harvested area (Monfreda et al. 2008) to generate baseline distribution maps circa 2000 of fertilizer
and manure application in cropland. The crop-area-based average N fertilizer and manure rates in
each grid cell (at a resolution of 5-arcmin) were calculated as:
$$\overline{C_{fer/man}} = \frac{\sum_i(C_{fer/man,i} \times AH_i)}{\sum_i AH_i} \tag{1}$$

where $\overline{C_{fer/man}}$ is the area-weighted average of N fertilizer or manure application rates (i.e.,
gridded baseline fertilizer or manure application rate, in the unit of g N m$^{-2}$ cropland yr$^{-1}$).
$C_{fer/man,i}$ and $AH_i$ are crop-specific N fertilizer or manure application rate (g N m$^{-2}$) and
harvested area (m$^2$), respectively, for crop type $i$.





Second, we used annual country-level N fertilizer and manure application amounts from FAO
(1961-2019) and the annual cropland area to scale the baseline year 2000 maps of N fertilizer and
manure application rates across time using the following equation:
$$R_{fer/man,y,j} = \frac{FAO_{fer/man,y,j}}{\sum_{g=1}^{g=n \ in \ country \ j}(\overline{C_{fer/man}} \times AC_{y,g})}$$
(2)

where $R_{fer/man,y,j}$ is the regulation ratio (unitless) in the year $y$ and country $j$. $FAO_{fer/man,y,j}$ is
country-level total N fertilizer or manure use amount (g N yr⁻¹) on cropland derived from
FAOSTAT. $AC_{y,g}$ is the area of cropland (m²) derived from the historical land use data in the year
$y$ and grid $g$. The actual N fertilizer and manure application rates were then calculated using the
following equation:
$$N_{fer/man} = \overline{C_{fer/man}} \times R_{fer/man,y}$$
(3)

where $N_{fer,man}$ is the "real" gridded N fertilizer or manure use rates (g N m⁻² cropland yr⁻¹) in
the year $y$.
Then, we extended the fertilizer data back to 1925 and manure data back to 1860 using the global
N flux change rates (Holland et al. 2005). Since industrial production of synthetic fertilizer was
developed in the early 1910s, we further extend fertilizer data back by assuming the fertilizer
production linearly increased from 1910 to 1925. Finally, N fertilizer application in cropland was
further divided into the $NH_4^+$ form the $NO_3^-$ form based on the annual country-level $NH_4^+$
application ratio in total N fertilizer provided by Nishina et al. (2017). This data was estimated
based on FAOSTAT's consumption data by chemical fertilizer type, which takes into account the
$NH_4^+$ and $NO_3^-$ content in each fertilizer type individually.



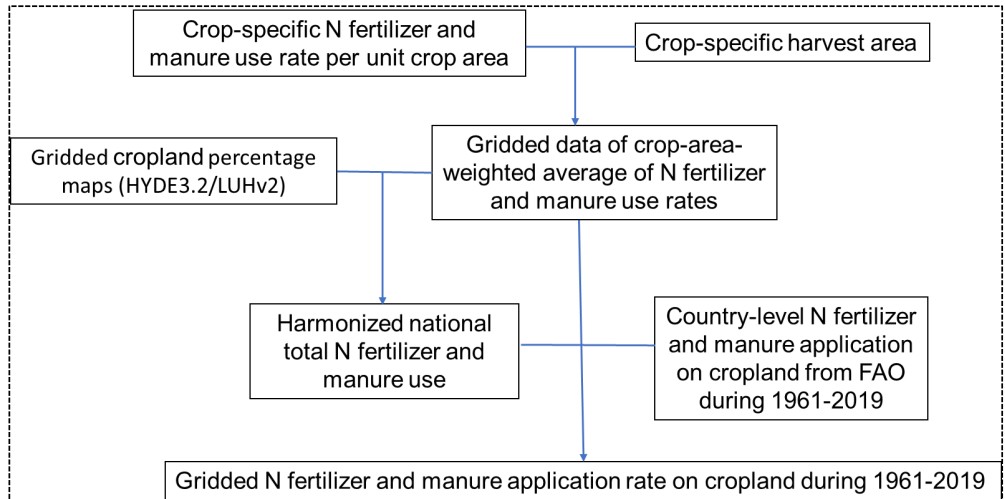

**Figure 1.** The workflow for developing the dataset of global annual N fertilizer and manure application rates during 1961-2019.

## 2.4. Spatializing the total manure N in pasture and rangeland

### 2.4.1. N fertilizer use in pasture

Due to the lack of grid-level spatial information of N fertilizer use in pasture, we assumed that pasture within each country has an even annual N fertilizer use rate. The fertilizer use in pasture per country was divided by the total pasture area of that country. Then this N fertilizer use rate per country was assigned to all the pasture grid cells in that country (Fig 2). The detailed method was introduced in Xu et al. (2019b).



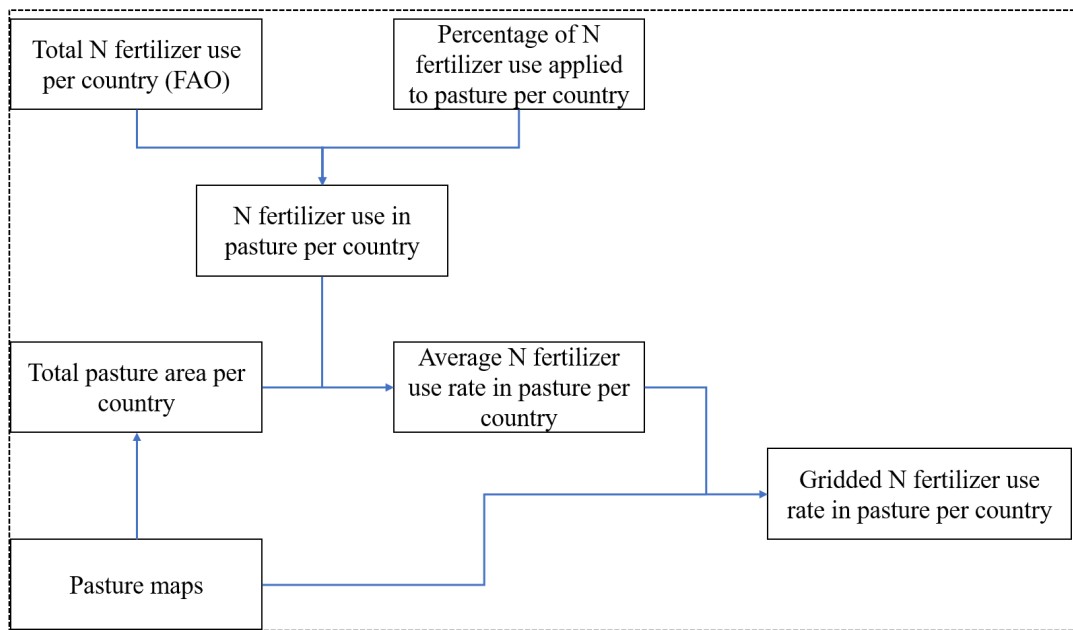


**Figure 2.** The workflow for developing the global pasture fertilizer application rate data during

208 1961-2019.

### 2.4.2. Spatializing manure application in pasture

To generate spatial patterns of manure application in pasture, we first calculated the spatial
distribution of annual manure N production. The Global Livestock of World 3 database (GLW3;
Gilbert et al., 2018) was used as a reference map of livestock distribution, which provided spatial
information for buffaloes, cattle, chickens, ducks, horses, goats, pigs, and sheep at a spatial
resolution of 0.083° in 2010. For the period 1961-2019, the FAO statistics of livestock population
in a country in one year was compared with the sum of GLW3 grid values within that country and
the ratio of the two values was used to scale all the GLW3 grid values of the country to generate
the spatial distribution of livestock in that year (Fig. 3). This routine can be represented as:

$$D_{l,c,y}^{FAO} = D_{l,c}^{GLW3} \times \frac{T_{l,c,y}^{FAO}}{T_{l,c}^{GLW3}} \qquad (4)$$

where $T_{l,c,y}^{FAO}$ indicates the FAO statistics of the population of the $l$th type of livestock of country $c$
in year $y$, $T_{l,c}^{GLW3}$ indicates the national population of the $l$th type of livestock of country $c$
summarized from GLW3, $D_{l,c}^{GLW3}$ is the spatial distribution corresponding to $T_{l,c}^{GLW3}$, and $D_{l,c,y}^{FAO}$ is





the corresponding spatial distribution to $T_{l,c,y}^{FAO}$. Applying the IPCC Tier 1 methodology for N
excretion (Dong et al., 2006) to these derived spatial distribution maps of livestock, we can have
the spatial maps of annual manure production during 1961-2019. Specifically, the average daily N
excretion rate was different for each livestock and for each group of countries, which were
classified by socioeconomic and geographic conditions. All manure production data were
resampled to 5-arcmin to be consistent with the pasture land use data.
Manure application to pasture during 1961-2019 is then estimated using manure production and
pasture area (Fig. 3) as:
$$R_{c,y}^{Nprod/Napp} = \frac{sum(GNprod_{c,y}^{FAO} \times GParea_{c,y}^{LU})}{Napp_{c,y}^{FAO}}$$
(5)

$$GNapp_{c,y}^{FAO} = mask(R_{c,y}^{Nprod/Napp} \times GNprod_{c,y}^{FAO}, GParea_{c,y}^{LU})$$
(6)

where $R_{c,y}^{Nprod/Napp}$ is the ratio of N production over FAO manure application to pasture in
country $c$ in year $y$, $GNprod_{c,y}^{FAO}$ is the gridded manure production in country c in year y estimated
based on FAO statistics of livestock data, $GParea_{c,y}^{LU}$ is the gridded pasture area in country $c$ in
year $y$ from our land use data, and $GNapp_{c,y}^{FAO}$ is the corresponding gridded manure application to
pasture in country $c$ in year $y$ through masking the product raster of $R_{c,y}^{Nprod/Napp}$ and $GNprod_{c,y}^{FAO}$
by the $GParea_{c,y}^{LU}$ raster. The manure application to pasture in year $y$ during 1860-1960 was
estimated as the product of $GNprod_{c,y}^{Holland}$ and $R_{c,1961}^{Nprod/Napp}$ (Fig 3). As for the period 1860-1960,
the time series of manure application data were also generated according to the manure N change
rates derived from Holland et al. (2005).



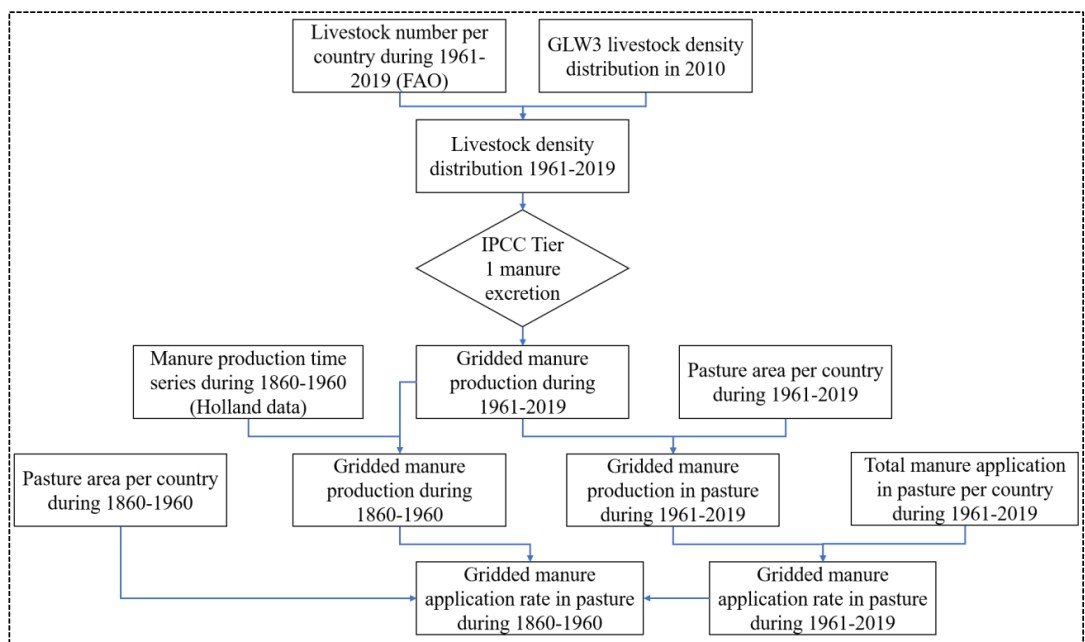

**Figure 3.** The workflow for developing the global pasture manure application rate data during 1860-2019.

### 2.4.3. Spatializing manure deposition in pasture and rangeland

The routine for spatializing FAO statistics of manure deposition on pasture and rangeland was similar to the method for manure application to pasture in Xu et al. (2019b). The only difference is that the manure deposition intensity on pasture was assumed to be twice that on rangeland within a grid cell, according to previous research (Campbell and Stafford Smith, 2000).

### 2.5 Atmospheric nitrogen deposition

Monthly atmospheric N depositions (NH$x$-N and NO$y$-N) during 1850–2014 were from N deposition fields of model simulations in the International Global Atmospheric Chemistry (IGAC)/Stratospheric Processes and Their Role in Climate (SPARC) Chemistry–Climate Model Initiative (CCMI) (Morgenstern et al., 2017). For the period 2015-2020, N deposition under SSP585 was used, consistent with TRENDY simulations for the global carbon budget (Friedlingstein et al., 2020). The CCMI models considered N emissions from multiple sources, including anthropogenic and biofuel sources, natural biogenic sources, biomass burning and lightning, and the transport of N gases and wet/dry N deposition (Eyring et al., 2013). The CCMI



N deposition data was developed in support of the Coupled Model Intercomparison Project Phase
6 (CMIP6) and used as the official products for CMIP6 models that lack interactive chemistry
components. The nearest interpolation method was used to resample N deposition data to a spatial
resolution of 5-arcmin.
**2.6 Regional Analysis**
In order to compare anthropogenic N inputs across different regions, we divided the global land
area into 18 regions according to national or continental boundaries (Tian et al. 2019). The 18
regions are USA, Canada (CAN), Central America (CAM), Northern South America (NSA),
Brazil (BRA), Southwest South America (SSA), Europe (EU), Northern Africa (NAF), Equatorial
Africa (EQAF), Southern Africa (SAF), Russia (RUS), Central Asia (CAS), Middle East (MIDE),
China (CHN), Korea and Japan (KAJ), South Asia (SAS), Southeast Asia (SEAS), and Oceania
(OCE).
**3. Results**
**3.1. Temporal and spatial changes in total anthropogenic N inputs**
The total anthropogenic N (TN) inputs to global terrestrial ecosystems increased from 29.05 Tg N
yr$^{-1}$ in the 1860s to 267.23 Tg N yr$^{-1}$ in the 2010s (Fig 4 and Table 2). The most rapid increase of
total N inputs was 3.53 Tg N yr$^{-2}$ occurred during 1945-1990 driven by both elevated fertilizer
application rates and cropland expansion. The TN inputs leveled off within the 1990s, but
increased again after 2001 with a lower increasing rate though. The TN inputs were dominated by
atmospheric N deposition before the 1900s. Manure N kept an increasing trend, accounting for
more than half of the TN inputs from the 1910s to the 1960s. Thereafter, the proportion of N
fertilizer substantially increased from 15% in the 1960s to 39 % in the 2010s, when manure N and
atmospheric N deposition accounted for 37% and 24% of N inputs, respectively.
Table 2. Decadal average of N inputs into the terrestrial ecosystem (Tg N yr$^{-1}$)

| Decade | Nfer NH$_4$ Crop | Nfer NO$_3$ Crop | Nfer NH$_4$ Pas | Nfer NO$_3$ Pas | Nman App Crop | Nman App Pas | Nman Dep Pas | Nman Dep Ran | Ndep NH$_x$ | Ndep NO$_y$ | Total |
|---|---|---|---|---|---|---|---|---|---|---|---|
| 1860s | 0.00 | 0.00 | 0.00 | 0.00 | 2.52 | 1.01 | 3.92 | 2.04 | 10.32 | 9.24 | 29.05 |
| 1910s | 0.08 | 0.05 | 0.00 | 0.00 | 6.54 | 2.20 | 9.87 | 6.38 | 11.59 | 10.72 | 47.43 |
| 1960s | 11.81 | 5.98 | 0.19 | 0.12 | 14.86 | 3.60 | 26.99 | 20.77 | 20.15 | 18.35 | 122.80 |
| 1970s | 28.21 | 12.09 | 1.21 | 0.72 | 17.23 | 4.14 | 30.77 | 23.17 | 25.40 | 22.98 | 165.94 |
| 1980s | 47.27 | 16.98 | 2.97 | 1.67 | 19.46 | 4.54 | 34.22 | 24.49 | 31.90 | 27.34 | 210.83 |




| | | | | | | | | | | | |
|---|---|---|---|---|---|---|---|---|---|---|---|
| 1990s | 56.42 | 14.59 | 4.02 | 1.73 | 20.19 | 4.29 | 36.99 | 25.67 | 33.80 | 28.55 | 226.26 |
| 2000s | 70.32 | 10.57 | 5.77 | 1.33 | 20.66 | 4.01 | 39.57 | 27.50 | 33.45 | 28.73 | 241.91 |
| 2010s | 87.52 | 9.03 | 7.39 | 1.10 | 22.29 | 4.09 | 43.25 | 28.68 | 35.58 | 28.30 | 267.23 |

Note: Nfer—N fertilizer, Nman—manure N, Ndep—N deposition, Crop—Cropland,
Pas—Pasture, Ran—Rangeland, App—Application, Dep—Deposition.

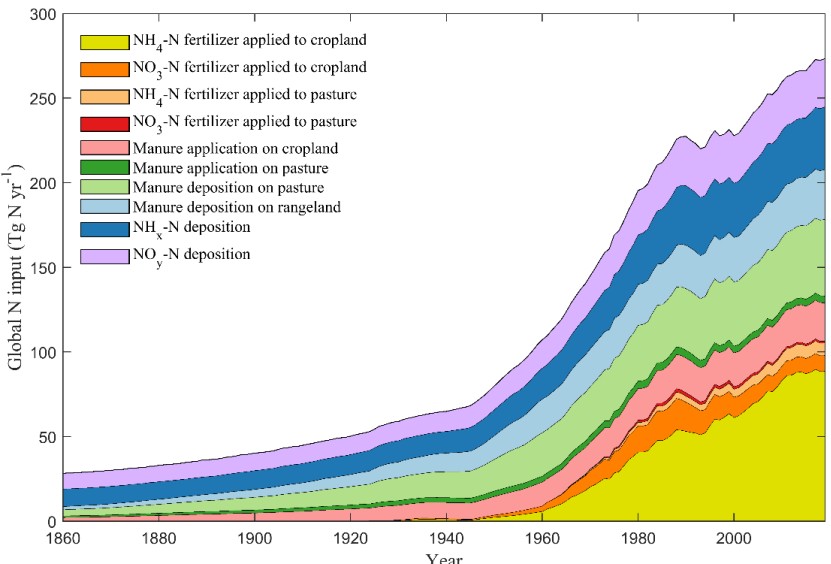


**Figure 4.** Long-term trends of anthropogenic nitrogen inputs to terrestrial ecosystems during 1860-2019. N input to global terrestrial ecosystems from three major categories: N fertilizer, manure N, and N deposition, which are further divided into ten specific types, including $NH_4$-N fertilizer applied to cropland, $NO_3$-N fertilizer applied to cropland, $NH_4$-N fertilizer applied to pasture, $NO_3$-N fertilizer applied to pasture, manure N application on cropland, manure N application on pasture, manure N deposition on pasture, manure N deposition on rangeland, $NH_x$-N deposition, and $NO_y$-N deposition.


The TN inputs exhibited high spatial heterogeneity across the globe, associated with the
imbalances in regional economic development and population growth (Fig. 5). From the 1860s to
the 1910s, the TN inputs mainly increased in the eastern U.S., Europe, and India, driven by the
increase in manure N application and deposition. In the 1960s, several hotspots of the TN inputs
emerged in Europe (Fig. 5c) where synthetic fertilizer was first widely used. Meanwhile, the TN
inputs were also intensified in many regions of the developing countries, such as eastern China,
southern Brazil, India, and countries in central Africa, mainly due to the increasing use of manure
N (Fig. 5c). As the access to the synthetic N fertilizer became easier, the TN inputs significantly
increased across the globe from the 1960s to the 2010s, and the inter-regional imbalance of N
inputs had also been amplified, with regions of high N inputs concentrated in eastern and central
China, India, Europe, midwestern U.S., and southern Brazil (Fig. 5d).

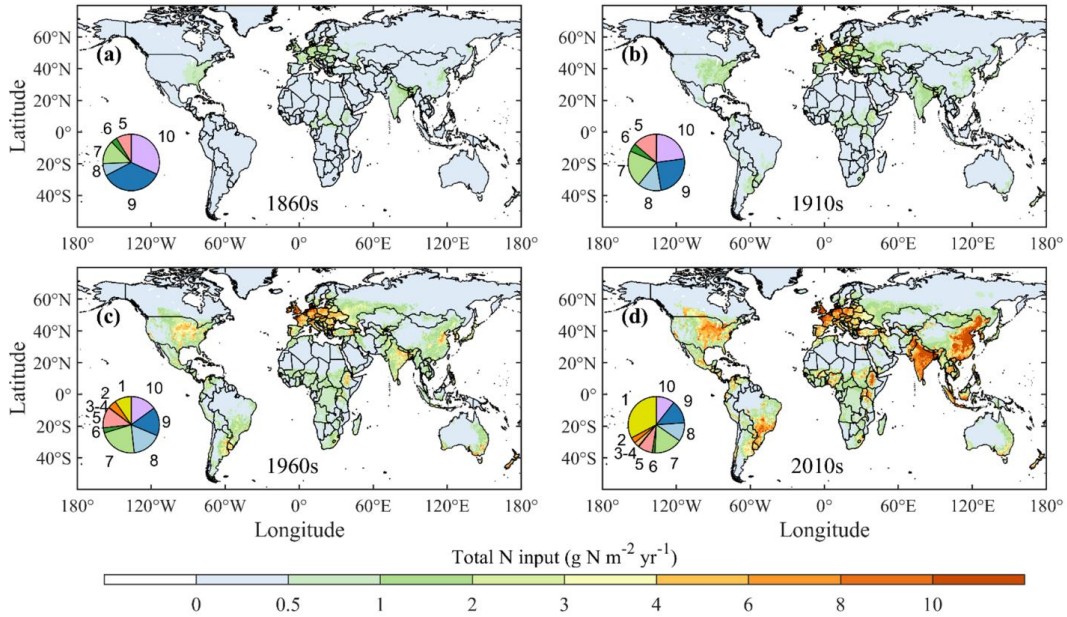


**Figure 5.** Spatial patterns of total N input in the (a) 1860s, (b) 1910s, (c) 1960s, and (d) 2010s.
For the pie chart in the spatial map, the numbers 1-10 represent the percentage of each component,
respectively (1. 'NH4-N fertilizer applied to cropland', 2. 'NO3-N fertilizer applied to cropland',
3. 'NH4-N fertilizer applied to pasture', 4. 'NO3-N fertilizer applied to pasture', 5. 'Manure
application on cropland', 6. 'Manure application on pasture', 7. 'Manure deposition on pasture',
8. 'Manure deposition on rangeland', 9. 'NHx-N deposition', 10. 'NOy-N deposition').


Among the 18 regions (Fig 6), the top three regions with the highest TN inputs in 1960 were
Europe (19.0 Tg N yr$^{-1}$), USA (11.8 Tg N yr$^{-1}$), and South Asia (9.9 Tg N yr$^{-1}$). From 1960 to 2019,
the largest increases in TN inputs were found in China, South Asia, and Brazil, which accounted
for 26%, 18%, and 9% of the increase of the global N inputs, respectively. The increasing TN
inputs in China and South Asia were mainly driven by the wide use of synthetic fertilizer, while
those in Brazil were driven by the use of both livestock manure and synthetic fertilizer. The TN



inputs in USA became relatively stable since 1980, whereas the TN inputs in Europe decreased by 32% from 1988 to 2019, primarily due to the increase in crop N use efficiency and the reduction in synthetic fertilizer application. Although the TN inputs in China experienced a rapid increase in recent decades, it started to show a decreasing trend after 2014. However, the TN inputs in South Asia and Brazil continued maintaining a strong growth trend. In 2019, China (49.1 Tg N yr$^{-1}$) contributed the largest share (18%) to global TN inputs, followed by South Asia (38.9 Tg N yr$^{-1}$, 14%) and Europe (26.2 Tg N yr$^{-1}$. 10%). The TN inputs in North America (USA and CAN), Europe (EU), East and South Asia (CHN, KAJ, SAS, and SFAS) were dominated by synthetic fertilizer, while those in Central and South America (BRA, SSA, NSA, and CAM), Africa (NAF, EQAF, and SAF), Central and West Asia (CAS and MIDE), and Oceania (OCE) were dominated by manure. RUS was the only region where atmospheric N deposition was the major anthropogenic N source in 2019.

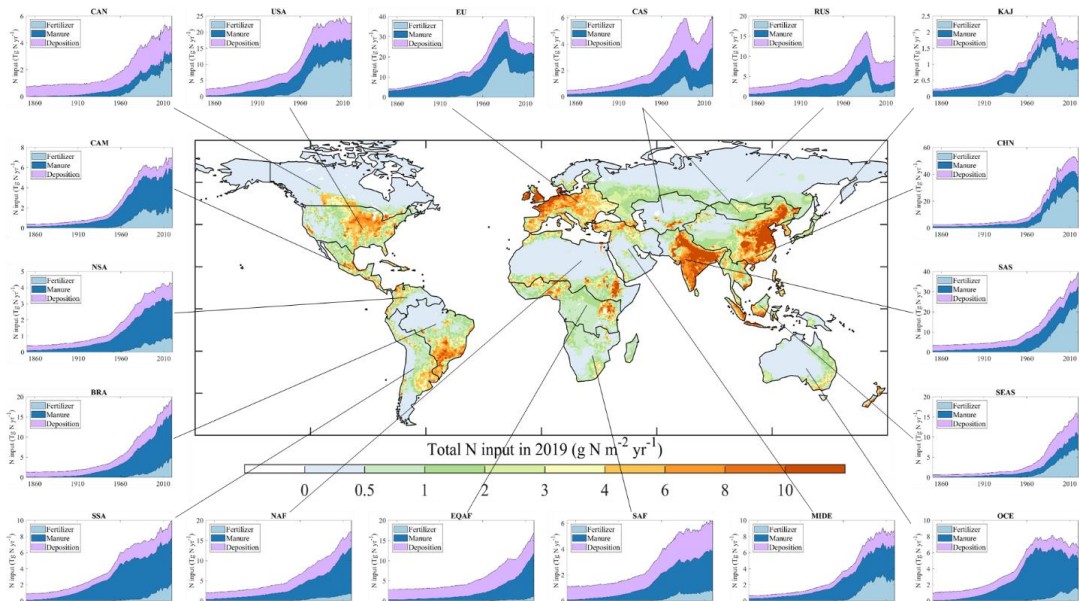

**Figure 6.** Long-term trends and variations of regional N inputs (synthetic fertilizer, livestock manure, and atmospheric deposition) to terrestrial ecosystems during 1860-2019. The 18 regions are USA, Canada (CAN), Central America (CAM), Northern South America (NSA), Brazil (BRA), Southwest South America (SSA), Europe (EU), Northern Africa (NAF), Equatorial Africa (EQAF), Southern Africa (SAF), Russia (RUS), Central Asia (CAS), Middle East (MIDE), China (CHN), Korea and Japan (KAJ), South Asia (SAS), Southeast Asia (SEAS), and Oceania (OCE).




### 3.2. N fertilizer inputs on cropland and pasture

From the 1960s to the 2010s, the N fertilizer inputs on cropland and pasture increased from 18.1
Tg N yr$^{-1}$ to 105.0 Tg N yr$^{-1}$. Specifically, N fertilizer inputs on cropland increased from 17.8 Tg
N yr$^{-1}$ to 96.6 Tg N yr$^{-1}$, and N fertilizer inputs on pasture increased from 0.3 Tg N yr$^{-1}$ to 8.5 Tg
N yr$^{-1}$ (Fig. 4 and Table 1). The proportion of $NH_4^+$ fertilizer in N fertilizer increased from 64%
in the 1960s to 90% in the 2010s, contrarily $NO_3^-$-N fertilizer decreased from 36% in the 1960s
to 10% in the 2010s. At the regional level, Europe and USA were the top two N fertilizer-
consuming regions in the 1960s, accounting for 38% and 25% of global N fertilizer application,
while China (28%) and South Asia (21%) were the top two in the 2010s (Fig. 6). Fertilizer
application rates in China and South Asia increased at a rate of 0.59 Tg N yr$^{-2}$ and 0.43 Tg N yr$^{-2}$
(p<0.05) during 1960-2019, respectively.

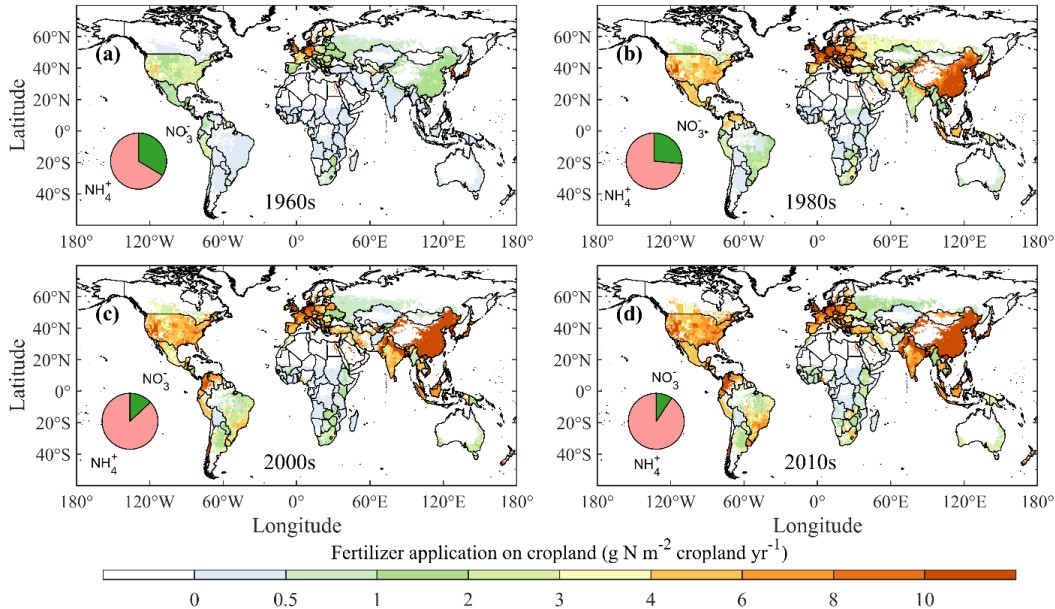


**Figure 7.** Spatial patterns of N fertilizer application on cropland in the 1960s, 1980s, 2000s, and
2010s.

Fertilizer application rates on cropland in Europe reached the maximum in the 1980s, but fertilizer
application rates in India, eastern Asia, and southern Brazil kept increasing continuously (Fig 7).
In the 2010s, extremely high N fertilizer inputs (> 20.0 g N m$^{-2}$ yr$^{-1}$) mainly occurred in eastern
and southeastern China. Croplands in northern India and western Europe also had high N fertilizer
rates (> 10.0 g N m$^{-2}$ yr$^{-1}$). N fertilizer application changed slowly in Africa, with most croplands
receiving N fertilizer less than 2.0 g N m$^{-2}$ yr$^{-1}$. For pasture, Europe was the main region with N
fertilizer application over 6.0 g N m$^{-2}$ yr$^{-1}$ before the 1980s (Fig 8). N fertilizer application on
pasture in southern Canada and India increased significantly with rates over 8.0 g N m$^{-2}$ yr$^{-1}$ in the
2010s. Most other regions (e.g., China, U.S. Brazil, Africa) received N fertilizer application of less
than 3.0 g N m$^{-2}$ yr$^{-1}$.

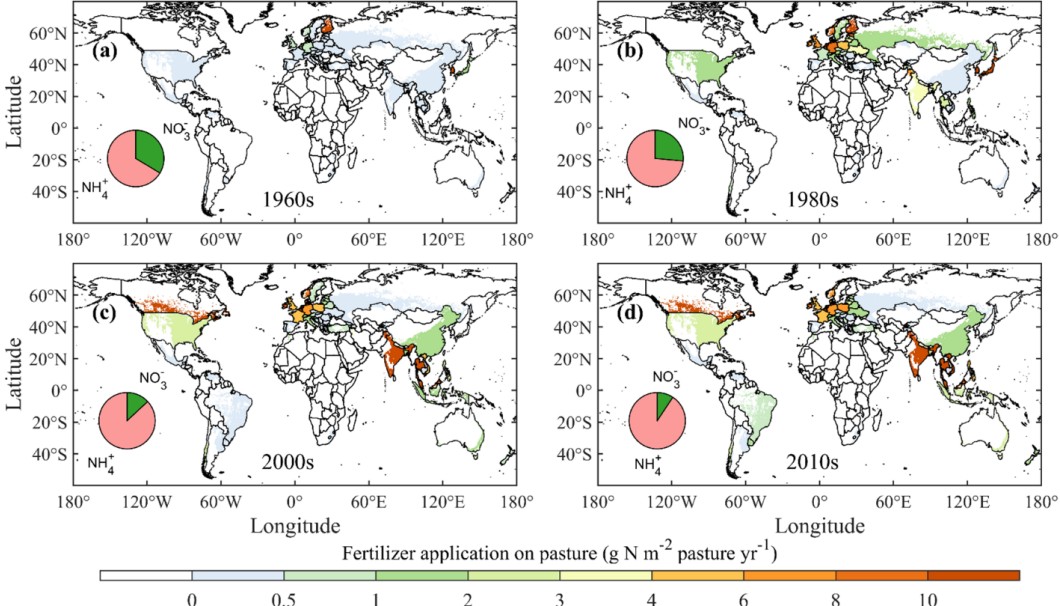

**Figure 8.** Spatial patterns of N fertilizer application on pasture in the 1960s, 1980s, 2000s, and
2010s.

**3.3. Manure N inputs on cropland, pasture, and rangeland**
The total manure N inputs to land increased from 9.48 Tg N yr$^{-1}$ in the 1860s to 98.31 Tg N yr$^{-1}$ in
the 2010s, with an increasing rate of 0.6 Tg N yr$^{-2}$ (Fig 4 and Table 1). The manure N application
on cropland, manure application on pasture, manure deposition on pasture, and manure deposition
on rangeland changed from 14.86 Tg N yr$^{-1}$ (22% of total manure input), 3.60 Tg N yr$^{-1}$ (5%),



26.99 Tg N yr⁻¹ (41%), and 20.77 Tg N yr⁻¹ (31%) in the 1960s to 22.29 Tg N yr⁻¹ (23%), 4.09 Tg
N yr⁻¹ (4%), 43.25 Tg N yr⁻¹ (44%), and 28.68 Tg N yr⁻¹ (29%) in the 2010s, respectively. Europe
was the largest contributor (39%) to global manure N inputs in the 1860s, but its share decreased
in the last century and became 9% in the 2010s (Fig. 6). The manure N inputs in Brazil grew
rapidly from 0.55 Tg N yr⁻¹ (2% of global manure N inputs) in the 1910s to 10.77 Tg N yr⁻¹ (11%)
in the 2010s. Similarly, manure N inputs in Equatorial Africa and Northern Africa were only 2.22
Tg N yr⁻¹ (3%) and 4.20 Tg N yr⁻¹ (6%) in the 1960s and increased dramatically to 9.40 Tg N yr⁻¹
(10%) and 10.60 Tg N yr⁻¹ (11%) in the 2010s, respectively. China was the largest contributor
(12%) of global total manure N inputs in the 2010s, while it contributed 8% in the 1960s and 12%
in the 1860s.

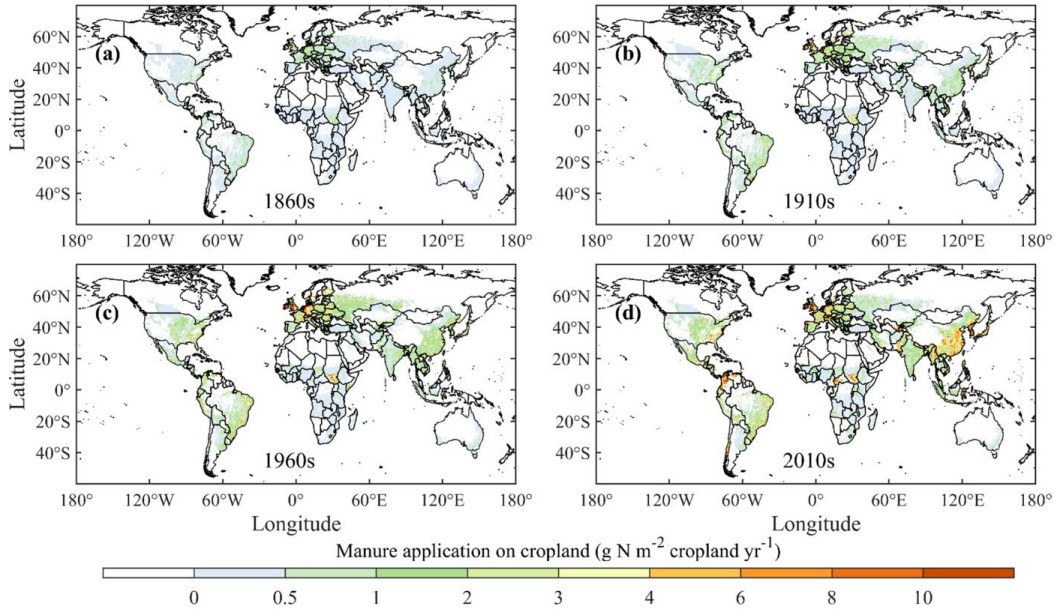


**Figure 9.** Spatial patterns of manure N application on cropland in the 1860s, 1910s, 1960s, and
2010s.

Manure application rates on cropland gradually intensified across the globe since the 1860s except
in Australia and part of Africa (Fig. 9). Hotspots of manure application on cropland (> 6.0 g N m⁻²
yr⁻¹) first appeared in western Europe in the 1910s, then intensified manure application was
observed in eastern Asia and northern South America in the 2010s. Manure application and



deposition on pasture had higher spatial variability than that on cropland (Fig. 10). Pasture in
Europe and South Asia received higher manure N than that in other regions. Eastern South America,
central Africa, and eastern Asia also experienced a significant increase in manure N inputs on
pasture since the 1910s. For manure deposition on rangeland, South Asia stood out over the study
period, with several other hotspots emerging in central Africa, northern China, Europe, and eastern
South America since the 1910s (Fig 11).

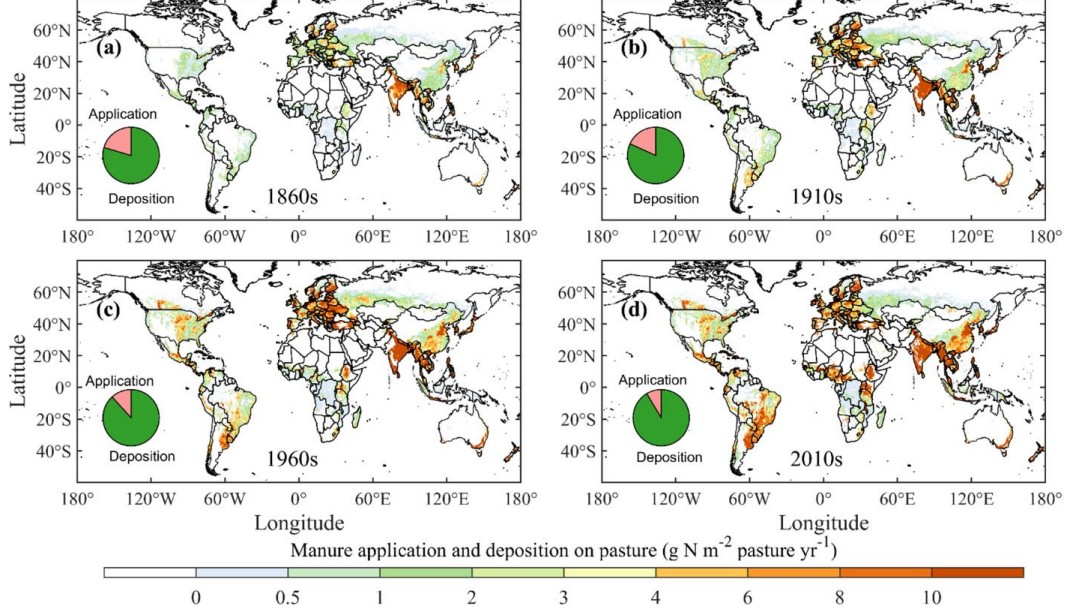


**Figure 10.** Spatial patterns of manure N application and deposition on pasture in the 1860s,
1910s, 1960s, and 2010s.

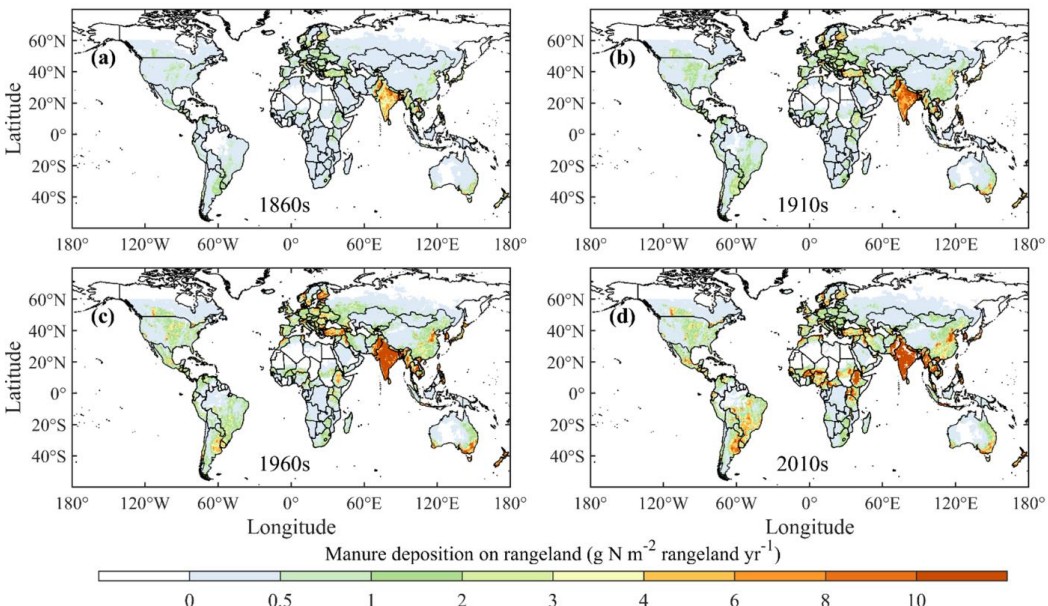


**Figure 11.** Spatial patterns of manure N deposition on rangeland in the 1860s, 1910s, 1960s, and 2010s.

### 3.4. Atmospheric N deposition on land

Atmospheric N deposition has a threefold increase from 19.06 Tg N yr$^{-1}$ to 60.87 Tg N yr$^{-1}$ during the 1850s - the 2010s, with NH$_x$ deposition increasing from 10.02 Tg N yr$^{-1}$ to 35.58 Tg N yr$^{-1}$ and NO$_y$ deposition increasing from 9.04 Tg N yr$^{-1}$ to 28.30 Tg N yr$^{-1}$ (Fig 4 and Table 1). The share of NH$_x$ in atmospheric N deposition started to increase after the 1970s, changing from 52% to 56% in the 2010s. At the regional scale, South Asia, Equatorial Africa, and USA were the largest contributors in the 1860s, accounting for 13%, 13%, and 12% of global atmospheric N deposition, respectively (Fig 6). In the 2010s, China was the region with the largest atmospheric N deposition (10.66 Tg N yr$^{-1}$, 17% of global atmospheric N deposition), followed by South Asia (5.90 Tg N yr$^{-1}$, 9%) and USA (5.69 Tg N yr$^{-1}$, 9%). Atmospheric N deposition peaked in the 1980s in Europe and Equatorial Africa, the 1990s in USA, and the 2010s in South Asia and China. Spatially, atmospheric N deposition intensified and increased dramatically across the globe since the 1910s (Fig. 12), and regions with high N deposition rates (>1.0 g N m$^{-2}$ yr$^{-1}$) were mainly in Europe, central Africa, southern Asia, U.S. (since the 1960s), and eastern Asia (in the 2010s).


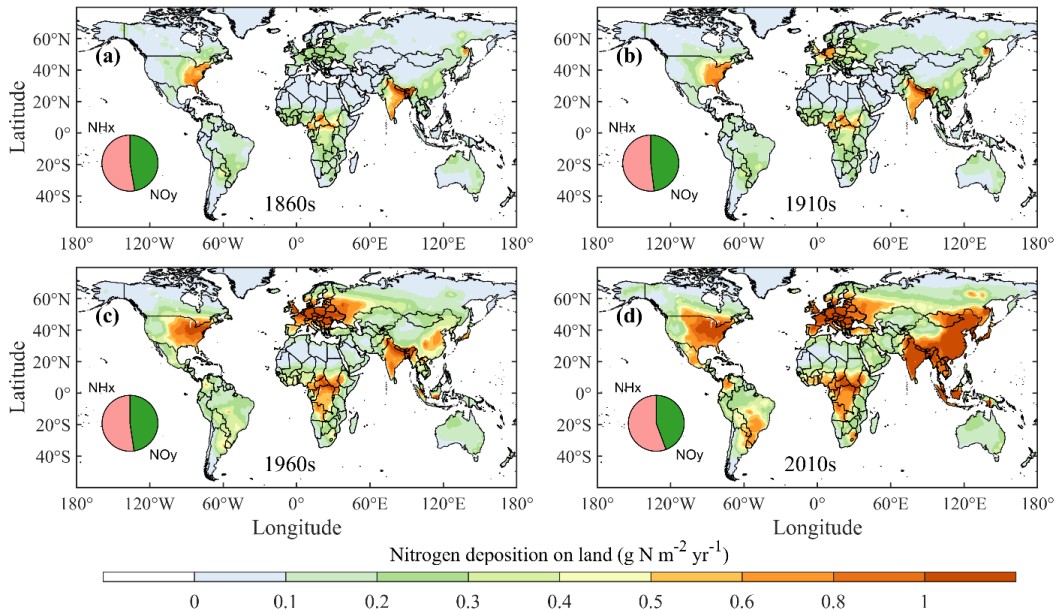

**Figure 12.** Spatial patterns of atmospheric N deposition on land in the 1860s, 1910s, 1960s, and 2010s.

## 4. Discussion

### 4.1 Socioeconomic forcing of N use

The total anthropogenic nitrogen inputs (excluding N deposition) showed a close relationship with GDP per capita in all the three agricultural sectors of cropland, pasture, and rangeland (Fig. 13). These relationships could be generally categorized into three groups: a hump-shaped curve, a rapid increase curve, and an asymptote curve. The first was typically seen in regions like China and Europe. China, as the top N consumer, has successfully reduced its nitrogen use for crop production from the peak of 33.6 Tg yr$^{-1}$ in 2014 to 30.0 Tg yr$^{-1}$ in 2020. Crop production in China increased in the same period due to the improvements in crop varieties, fertilizer management, and land use policies (Cui et al., 2018; Wu et al., 2018). The mandatory policies and directives for N use in Europe since the late 1980s have effectively curbed its N use to a stable level (Van Grinsven et al., 2014). The second could be seen in South Asia, Southeast Asia, North Africa, etc. These regions are still in the developing stage and need to tackle the food demand of rapidly growing population, which, together with low nitrogen use efficiency, results in a surge of nitrogen



pollution (Chang et al., 2021). The third could be well represented by USA and Canada. For the
USA, although its crop nitrogen use efficiency has considerably improved since the 1990s driven
by technological and management improvements (Zhang et al., 2015), its cropland area has kept
expanding recently with the new cropland usually producing yields below the national average
(Lark et al., 2020), which undermines its efforts for reducing N excess induced environmental
pollution. For the same curve type, there also existed obvious differences. For example, the turning
points for crop N inputs in Europe and China emerged at varied socioeconomic development levels.
Meanwhile, it was difficult to predict when China's crop N inputs would decrease to its lowest as
Europe's case had shown. For different sectors of one country or region, their N inputs could also
show asynchrony with GDP per capita increases. Take the USA as an instance, its N inputs on
cropland and pasture kept growing while its N inputs on rangeland had kept stable. Despite such a
diversity of the N use changes in varied socioeconomic circumstances, the N use-GDP per capita
relationships and the related spatial patterns will be a valuable reference for any future projection
of global anthropogenic N inputs.

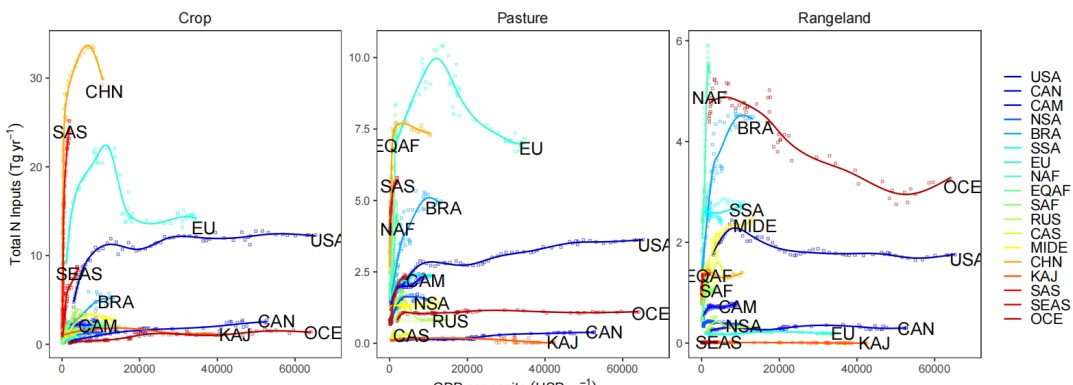

**Figure 13.** Relationships between total N inputs (excluding N deposition) and GDP per capita in
cropland, pasture and rangeland, respectively, within each of 18 regions during 1961-2019. The
lines were fitted using the generalized additive models. For displaying clarity, not all region names
are shown in each panel.

**4.2 Implications for nitrogen use management**
Excessive N use has induced a variety of environmental issues, due to the magnitude, trend and
the constitute forms. In regions or countries like Europe and the US, though the N inputs have been



stable (Fig. 13), the large magnitude of annual N inputs results in a considerable fraction of reactive N that is stored in soils. This N pool can cause strong legacy effects, of which the influence on water quality would last for decades (Meter et al., 2018). Therefore, maintaining the current levels of N inputs is far from reducing N related environmental issues in these regions or countries (Liu et al. 2016). Instead, agricultural nitrogen inputs are required to be eliminated drastically, which, however, seems rather difficult at the current technological level even the social-economic conditions are improving (Fig. 13). But for regions or countries like South Asia and Southeast Asia, where N inputs have been increasing rapidly, the management options or activities that are successful in Europe or USA can be promoted to inhibit the further increase of anthropogenic N inputs and local N induced pollution. This requires wide international collaboration and efficient coordination between developing countries and developed countries. As for the changes in N input forms, a signal worth noting is the increasing fraction $NH_4^+$-N in the global total N inputs (Figs. 7, 8, and 12). High $NH_4^+$-N fraction has contributed significantly to N induced air pollution (Li et al., 2016), and the change of the ratio of $NH_4^+$-N over $NO_3^-$-N may affect biodiversity (van den Berg et al., 2016) and plant growth (Zhu et al., 2020; Yan et al., 2019). Improved use of $NH_4^+$-N will benefit both human society and ecosystems.

**4.3 Limitations in data development and knowledge gaps**

The uncertainties and limitations of this global N input dataset are mainly derived from the following aspects: (1) Land use maps. Cropland, pasture, and rangeland distribution maps are critical for the spatialization of N fertilizer and manure application. In the data development process, we constrain N input amount of this dataset with the country-level fertilizer/manure consumption from FAO to ensure the total input consistent, but fertilizer use rate per unit cropland area could be significantly biased if the global data differs a lot from the country-specific data. For example, in the US, the higher cropland acreage in HYDE/LUH2 database, compared with the USDA census, is likely to make fertilizer input rate diluted, which could affect the impact assessment of N inputs (Yu and Lu, 2018). (2) Spatial patterns of fertilizer and manure application rate. The baseline of crop-specific fertilizer and manure use rates is fixed and has been used to determine the spatial patterns of fertilizer and manure inputs over the study period. This conflicts with the reality of inter-and intra-annual dynamics of crop rotation, annual changes in crop harvested area as well as changes in crop-specific fertilizer use rate over time. An ideal spatially explicit fertilizer input data, in the future, ought to consider the dynamics of crop rotation,





individual crop area changes, and crop-specific fertilizer use rate over space and time. (3) Country-
level survey data. The country-level fertilizer and manure data from FAO don't separate N
application to cropland and pasture. In this study, we separated fertilizer and manure application
to cropland and pasture simply based on constant ratios generated by Lassaletta et al. (2014) and
Zhang et al. (2015), which ignored either the temporal or the spatial changes of allocation of
fertilizer and manure application to cropland and pasture. (4) Pre-1961 N inputs. Since the country-
level fertilizer and manure data are only available after 1961, we assumed the change rates of
global manure and fertilizer inputs before 1961 followed the change rates of annual global data
reported by Holland et al. (2005). (5) Other N sources to terrestrial ecosystems. Leguminous green
manure, which performs biological N fixation, was the most common nitrogen-containing soil
fertility maintenance cropping practice before the widespread use of synthetic fertilizer, and is also
used in current organic farming practices (Cherr et al., 2006). Since there are no statistics on the
types and use of green manure on a global scale, it is necessary to develop a related database in
future.
For future data improvements, we call for advanced N management survey/reporting mechanism
to develop fine-scale N consumption or use rate data. For example, the commonly used survey
data for the global fertilizer database is country-level consumption amount or crop-specific
fertilizer input from IFA and FAO, which smoothed large variations in fertilizer application rate
at farm level and sub-national scales. A continuous survey of crop-specific fertilizer and manure
use at sub-national scale, development of dynamic global land use data, and crop rotation maps
with more precise regional patterns are important for improving the resolution and accuracy of
geospatial fertilizer and manure data. Additionally, considering fertilizer and manure application
timing in the data is also important for agricultural nutrient management, which relies on the efforts
and investigations regarding the fertilizer and manure application behavior at multiple spatial
scales.
**Data availability**
The History of Anthropogenic N Inputs (HaNi) dataset is available at
https://doi.pangaea.de/10.1594/PANGAEA.942069 (Tian et al., 2022).



**Summary**

In this work, we developed a global annual anthropogenic N input dataset at 5-arcmin resolution during 1860-2019 by integrating multiple available databases into a uniform framework. This dataset for characterizing the History of anthropogenic N inputs (HaNi) includes major pathways and species of anthropogenic N input to the terrestrial biosphere, such as synthetic fertilizer N use in cropland and pasture, manure N application in cropland and pasture, manure N deposition in pasture and rangeland, and atmospheric N deposition. The TN input to global terrestrial ecosystems raised rapidly since the 1940s due to the widespread usage of synthetic N fertilizer, and the increase started to slow down after 2010. The hotpots of TN inputs shifted from Europe and North America to eastern and southern Asia. The TN inputs in North America, Europe, and East and South Asia were dominated by synthetic fertilizer, while those in Central and South America, Africa, Central, and West Asia, and Oceania were dominated by livestock manure. The N usage varied significantly in different socioeconomic circumstances, but the N use-GDP relationships still could provide a valuable reference for future projection of global anthropogenic N inputs. The HaNi dataset can serve as input data for a wide variety of modeling studies in earth system and its components (land, water, atmosphere and ocean), providing detailed information for the assessment of anthropogenic N enrichment impacts on global N cycling and cascading effects on climate, ecosystem, air and water quality. This data will keep updated in the future.

**Author contributions**

H.T. designed and led this work. Z.B., H.S. and X.Q. were responsible for developing the datasets. N.P. plotted all figures. F.N.T. and G.C. provided the FAO dataset. N.M. provided the crop-specific fertilizer and manure datasets. K.N. provided the fertilizer type dataset. S.P., C.L., and R.X. proposed the methods in the study. All authors contributed to the writing of the manuscript.

**Competing interests**

The authors declare that they have no conflict of interest.



**Acknowledgments**

This study has been partly supported by National Science Foundation (Grant numbers: 1903722; 1922687), Andrew Carnegie fellowship Program (Grant No. G-F-19-56910). J.L. acknowledge National Natural Science Foundation of China (Grant no. 41625001), H.S. and X.Q. acknowledge National Key R & D Program of China (grant nos. 2017YFA0604702) and National Natural Science Foundation of China (Grant no. 41961124006). F.N.T. acknowledges funding from FAO regular programme. The views expressed in this publication are those of the author(s) and do not necessarily reflect the views or policies of FAO. K.N. is supported by a project, JPNP18016, commissioned by the New Energy and Industrial Technology Development Organization (NEDO) and a JSPS KAKENHI Grant Number JP18K11672. J.C. is supported by the Fundamental Research Funds for the Central Universities (2021QNA6005). We also thank James Gerber for providing us with the data on crop-specific nitrogen manure application.



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
