# Peer review of "A 5-arcmin resolution annual dataset from 1860 to 2019"

_Earth System Science Data, 2022_

## Author Comment (AC1)

Earth System Science Data (ESSD)

Sep 17, 2022

Dear Editor and reviewers:

Thank you for your efforts and comments concerning our manuscript entitled "History of anthropogenic Nitrogen inputs (HaNi) to the terrestrial biosphere: A 5-arcmin resolution annual dataset from 1860 to 2019" (ID: essd-2022-94). These constructive comments are valuable and helpful for improving our manuscript, as well as providing important guidance to our future research. We have addressed all comments raised by reviewers and provided point-to-point responses outlined in the Response to Reviewers enclosed. Hope the corrections and explanations we made can meet with your approval.

Thanks in advance for your consideration.

Sincerely,

Dr. Hanqin Tian

Institute Professor of Global Sustainability, Schiller Institute for Integrated Science and Society [1]

Professor, Department of Earth and Environmental Sciences

Boston College, 245 Beacon St., Chestnut Hill, MA 02467, USA; E-mail: hanqin.tian@bc.edu; Phone: (617) 552-3664

**Responses to the Reviewers' comments**

**Reviewer 1**

This study is a very impressive work that comprehensively reconstructs the History of anthropogenic N inputs to the terrestrial biosphere, especially with consistent temporal coverage, spatial resolution, and spatial allocation. This work is beneficial for a comprehensive assessment of the coupled human-earth system and addresses a variety of social welfare issues, including climate-biosphere feedback, air pollution, water quality, and biodiversity. Although uncertainties and limitations occur in this global N input dataset, I think this work has taken a big step forward anyway. This paper is reasonably well written. I fully support the submission of this paper with minor revisions. The detailed comments are as follows:

**Comment 1**: Line 200, the subtitle is not quite correct, since in this part you also describe the method for N fertilizer use in the pasture.

Response: Thanks for your careful check. The subtitle has been revised as "Spatializing the total fertilizer and manure N in pasture and rangeland". **L206.**

**Comment 2**: Line 210-241 Does the method of spatializing manure application in pasture differ from that in Xu et al. (2019)? Can you explain where the improvements are made? I must be understanding something wrong, otherwise, I can't figure out why the ratios of the two methods are exactly opposite. What's the definition of Napp_c,y, and how to get the value of Napp_c,y. In addition, the GNprod_c,y you get should be adjusted and compared with the national-level application amounts from the FAOSTAT database.

Response: The reviewer's understanding is correct that the method of spatializing manure application in pasture is the same as Xu et al. (2019). The difference lies in that they used gridded manure production from Zhang et al. (2017), in which the Global Livestock of World 2 database (GLW2) was used and the animals considered were cattle, swine, chickens, goats, sheep, and ducks, whereas we used the GLW3 dataset and considered buffaloes and horses additionally. The advantage of taking into account more animals was that the gap between our estimate of global manure production in 1961 and that by Holland et al. (2005) in 1960 was largely reduced. The ratios of manure application over manure production in our work and theirs, as the reviewer expected, were in the same format, and here we apologize for incorrectly presenting the ratio equation. The original wrong equation has been revised as its reciprocal (see Eqs. 5 & 6 in main text) and we checked our code thoroughly and found no mistake. Moreover, we clarified the definition of $Napp_{c,y}^{FAO}$ which indicates FAO statistics of manure application to

pasture in country $c$ in year $y$. As for the gridded manure production in a country $c$ in year $y$, i.e., $GNprod_{c,y}^{FAO}$, it was estimated using IPCC Tier 1 manure excretion algorithms based on FAO statistics of livestock number (Fig. 3), while FAO itself does not provide such an estimation.

**Comment 3**: Line 224, the reference of Dong et al., 2006 can not be found in the reference list. Please check throughout the manuscript.

Response: The reference (Dong, H., Mangino, J., Mcallister, T. A., Hatfield, J. L., Johnson, D. E., Lassey, K. R., de Lima, M. A., and Romanovskaya, A.: Chapter 10: emissions from livestock and manure management, in: IPCC guidelines for national greenhouse gas inventories, 4, 2006.) has been added and we checked all citations thoroughly. **L230.**

**Comment 4**: Line 247-248. The method you used for spatializing manure deposition in pasture and rangeland is similar to the method for manure deposition to pasture in Xu et al., not the manure application method.

Response: The word "application" has been changed to "deposition". **L255.**

**Comment 5**: Line 255-256, what's the full name of SSP585 and TRENDY. A full name is required on the first occurrence. Please check throughout the manuscript.

Response: Thanks for your reminding us. "The highest emission scenario in shared socio-economic pathways" is added to explain SSP585 and "Dynamic Global Vegetation Model simulations" is added to explain TRENDY. **L270-271.**

**Comment 6**: Line 280-282, when manure N and atmospheric N deposition accounted for 37% and 24% of N inputs refer to which year? The sentence is not very clear and easy to cause ambiguity. Please rephrase it.

Response: The 37% and 24% refer to the 2010s. We rephrase this sentence as "the proportion of N fertilizer in TN inputs substantially increased from 15% in the 1960s to 39 % in the 2010s, meanwhile, the proportions of manure N and atmospheric N deposition decreased from 54% and 31% to 37% and 24%, respectively". **L296-298.**

**Comment 7**: You might cite and compare with the recent work of Wang et al. (published in National Science Review, 7: 441-452, 2020), which developed crop-specific N fertilizer inputs applied to cropland from 1961-2014.

Response: Thanks for recommending this related study. We have cited this paper in the introduction. **L83.**

**Reviewer 2**

The manuscript entitled "History of anthropogenic Nitrogen inputs (HaNi) to the terrestrial biosphere: A 5-arcmin resolution annual dataset from 1860 to 2019" and its associated dataset represent an impressive and very relevant contribution to the scientific understanding of the global history of N cycle. The analysis is robust and the manuscript is well written and structured. The dataset will surely be very useful for other scientists, even if it has a number of limitations that should be taken into account. Therefore, I recommend its publication after some minor comments have been addressed.

Most of the main problems I have found are already discussed by the authors in the "Limitations in data development and knowledge gaps" section. Therefore, I will not repeat them, but I will rather focus on issues that have not been mentioned by the authors. In some cases, I provide suggestions for improvement. If the authors do not want to change their methods (in fact I think that the quality of the dataset is already good enough to be published) they should at least justify their choices.

General comments

**Comment 8**: It is a pity that you don't include BNF, although I understand the reasons. In fact, the problem of data lack at the global scale is not only related to information on cover crops, but also on weeds and on legume crops themselves. In any case, I think that including legume crops BNF would have been better than nothing…

Response: The reviewer's comment is great! We also realized this BNF issue but finally we determined not to include it for two reasons: 1) the estimate of agricultural BNF needs long-term spatial and temporal distribution maps of various crops such as cover crops and legume crops, which are not available at the global scale for now to our knowledge. This means it would be better to treat the development of agricultural BNF data as independent work, as the involved methodology will be much different from that used in this study. In fact, we are now collecting BNF measurements for crops and developing remote sensing based crop distribution maps and hopefully this work can be completed this year. 2) the HaNi dataset was developed to serve as inputs for terrestrial biosphere models and thus the N components like BNF, which are simulated using different mechanisms in the models, were not included. But for sure an available agricultural BNF dataset will be extremely meaningful for being at least a benchmark for the models. In the Discussion section, we mentioned this BNF issue and provide the existing estimation of global legume crop BNF (**Lines 532-537**) and further included the above statements to further clarify why BNF is not considered (**Lines 537-544**).

**Comment 9**: The limitation number (2) that you mention, regarding the used of fixed spatial patterns of fertilizer and manure application, is critical. As you acknowledge, crop distribution and N rates have changed over time. But in addition, the spatial distribution of livestock, in particular, has greatly changed with industrialization, which would probably lead to changing spatial distributions of total manure over time, given that manure is not usually transported at large distances. This would affect the mean N rates in each grid cell.

Response: Thanks for pointing it out. The overlook of the change in livestock distribution is indeed one of the limitations of this dataset. We added your comment in the limitation (2). "In addition, the spatial distribution of livestock has greatly changed with industrialization, which would probably lead to changing spatial distributions of total manure over time, given that manure is not usually transported at large distances.". **L522-525.**

**Comment 10**: In addition, manure excretion rates have been considered fixed for each country, which is not realistic given the historical changes in livestock live weight and productivity over time (Zhang et al., 2022). I think that a more realistic approach would be to estimate livestock excretion based on intake and production, for example using the data from Zhang et al. (2022).

Response: Indeed, it is more realistic to estimate livestock excretion based on intake and feed digestibility, as proposed by the IPCC Tier 2 methodology. However, the Tier 2 method is not applicable to all types of animals due to parametrization difficulty, for example, pigs, as shown by Zhang et al. (2022). Meanwhile, to be consistent with the manure production before 1960 estimated by Holland *et al.* (2005), the IPCC Tier 1 method was adopted for this study. But we added extra statements in the Discussion section to mention this limit in the methodology. **L504-509.**

**Comment 11**: If I understood correctly, you estimated (1) manure application to cropland from FAOSTAT and Holland, (2) manure application to grassland based on the combination of your own estimation of livestock excretion with the application to grassland reported in FAOSTAT, and (3) manure deposition on grassland based on FAOSTAT and Zhang et al. (2017), following the method of Xu et al. (2019). I wonder if these three approaches have been harmonized in some way, to avoid inconsistencies in the estimation of total livestock excretion and unrealistic transport distances.

Response: We would like firstly to clarify that manure deposition (either to pasture or rangeland) was also based on FAOSTAT and manure production (i.e., manure execration), with the latter produced by harmonizing our own manure production estimates and manure production by Holland et al. (2005). The detailed processes were described in Section 2.4.3 and Fig. 3. To avoid inconsistencies between total manure use and total manure production within a grid cell and unrealistic transport distances, the sum of manure application to cropland and pasture and manure deposition to pasture and rangeland was constrained to be less than or equal to manure

production within a grid cell. In fact, the case that the total manure use surpassed the total manure production in a grid cell was rare. If there was, the four components, namely the manure application to cropland and pasture and manure deposition to pasture or rangeland, were scaled by multiplying the ratio of their sum over the total manure production within a grid cell. We have added this clarification to Section 2.4.3. **L257-264.**

**Comment 12**: You talk about "total anthropogenic inputs to global terrestrial ecosystems" but you are not studying forests, which are also terrestrial ecosystems receiving anthropogenic N inputs from N deposition. This comment would also apply to the title of the paper, which mentions the "terrestrial biosphere"

Response: The atmospheric N deposition in this study represents the N deposition on whole terrestrial ecosystems, including forests.

**Comment 13**: Similarly, you include N deposition from all sources, including natural biogenic sources (this leads to relatively high N deposition in the early period of your time series, being the main N input in the 19th century). However, in the paper you say that you are studying "anthropogenic inputs". Isn't this inconsistent?

Response: Thanks for pointing it out. The atmospheric N deposition data were derived from the CCMI models. We could not find any supporting information to distinguish between anthropogenic source and natural source N deposition based on this dataset. We may still decide to keep using the name of anthropogenic N inputs for this dataset, because it consists of the major anthropogenic N sources, such as synthetic fertilizer and livestock manure. We clarify this issue of the "anthropogenic N inputs" in the discussion to make it clear that the "anthropogenic N inputs" in this study actually do not exclude the natural source of atmospheric N deposition and do not include legume crop biological N fixation. **L533-535.**

Specific comments

**Comment 14**: L.60. Check grammar: "has been developed" instead of "have been developed"?

Response: Thanks for your careful check. The "has" is used instead. **L63.**

**Comment 15**: 64. I suggest to add "respectively" after "chemical fertilizer"

Response: The word "respectively" has been added. **L67.**

**Comment 16**: 76 "aerosols" instead of "aerosol"?

Response: The "aerosols" is used instead. **L79.**

**Comment 17**: L.76 I would clarify why being a precursor of aerosols is relevant (e.g. adding "pollutants", or similar)

Response: This sentence has been revised as "The global emission of ammonia (NH3), a major precursor of aerosols contributing to air pollution, had rapidly increased from 1.0 Tg N yr-1 in 1961 to 9.9 Tg yr-1 in 2010, mainly due to the wide use of N fertilizer". **L78-81.**

**Comment 18**: 99 Tian, 2017 is not present in the reference list

Response: This reference refers to the Lu and Tian (2017).

**Comment 19**: 344. You refer to Table 1, which does not exist.

Response: We are sorry for the missing table. The Table 1 has been added. **L153.**

Table 1. Summary of main data sources

| Data Source | Dataset | Reference |
|---|---|---|
| FAOSTAT | Annual country-level fertilizer and manure inputs to land from 1961 to 2019 | FAO (2021) |
| EARTHSTAT | Fertilizer and manure application rates for major crops | (Mueller et al., 2012) (West et al., 2014) |
| EARTHSTAT | Harvested area and yield for major crops | (Monfreda et al., 2008) |
| Hyde3.2/LUHv2 | Cropland, Pasture, and rangeland area from 1860 to 2019 | (Klein Goldewijk et al., 2017) (Hurtt et al., 2020) |
| Holland et al., 2005 | Global fertilizer and manure N from 1860 to 1960 | (Holland et al., 2005) |
| Nishina et al., 2017 | Annual NH4 and NO3 fraction in total fertilizer from 1961 to 2014 | (Nishina et al., 2017) |
| GLW3 | Livestock distribution maps | (Gilbert et al., 2018) |
| Eyring et al. 2013 | Monthly atmospheric N depositions (NHx-N and NOy-N) during 1850–2014 | (Eyring et al., 2013) |

**Comment 20**: L.140. I think the proper word is "negligible", not "neglectable"

Response: The "negligible" is used instead. **L144.**

**Comment 21**: 141. In case you want to consider them, Zhang et al. (2015) (see Supplementary Information) implement some refinements to the Lassaleta's estimation of fertilizer partitioning between cropland and grassland. In addition, Einarsson et al. (2021) make a more refined estimation for European countries

Response: Yes, Zhang et al. (2015) determined the percentage of the fertilizer used for pastures according to Lassaletta et al. (2014), with several refinements based on Conant et al. (2013). Zhang only mentioned the refinement in France not all European countries, and we will incorporate this partition ratio of France during next year's date updating.

**Comment 22**: 146. Einarsson et al. (2021) make a more refined estimation of manure partitioning between cropland and grassland for European countries

Response: Thanks for your suggestion. Einarsson et al. (2021) provided a very detailed dataset of the shares of manure to cropland and grassland for 26 present-day European countries. Unfortunately, the paper and dataset became accessible after we submitted this paper, we will use these updated research results during next year's data updating.

**Comment 23**: 163. How did you estimate N rates of the remaining crop types? Monfreda et al. (2008) report data on 175 crops, and many of the cells in its dataset contain a very low share of the 17 dominant crop types reported in the fertilizer rate studies

Response: We calculated the average N application per area in each grid cell based on the 17 dominant crop types. The total N application is calculated by multiplying the average application rate and cropland area in each grid. If the grid cell contains a very low share of the 17 dominant crop types, the N application rate on other crop types was represented by the average N application rate of the 17 dominant crop types.

**Comment 24**: 184. In the equation you write Nfer/man instead of Nfer,man

Response: We have changed the Nfer,man to Nfer/man. **L191.**

**Comment 25**: L.186. The data in Holland is global. I guess you assumed the same change rates for all countries?

Response: Yes, we assumed the same change rates for all countries. Currently, we do not have any other reference to inform the changes before 1960 for different countries, therefore, we assumed the same change rates for all countries.

**Comment 26**: Table 2. Given that there have been important changes in cropland and grassland areas in the studied period, it would be interesting to see these numbers also expressed per unit area. Also, showing the total values of N inputs in cropland, pasture and rangeland would be very interesting. Some of this additional information could be included as supplementary material if there is not enough space in the main body of the article.

Response:  We added Table S1 in supplementary material according to your suggestion.

**Comment 27**: Figure 4. I would suggest placing N deposition on the bottom of the graph and N fertilization on the top. This way, the specific contribution of each category would be better seen (given than in the current form, all components of the graph take the shape of synthetic inputs).

Response: The figure has been revised according to your suggestion.

**Comment 28**: 321. A citation would be welcome to support the statement "primarily due to the increase in crop N use efficiency"

Response: The citation (Zhang et al., 2021a; Lassaletta et al., 2014) is added to support the increase in crop N use efficiency in US and Europe. **L339.**

**Comment 29**: Figure 6. A slight increase in the font size of the regional panels would facilitate the reading of the figure

Response: The figure has been revised according to your suggestion.

**Comment 30**: 344. Table 1 does not exist. I think you mean Table 2.

Response: Yes, it is Table 2, we have revised it. **L361.**

**Comment 31**: 349 I think you mean Tg N yr-1, not Tg N yr-2

Response: Usually for the amount of N input, we use the unit Tg N /yr, and for the change rate of the amount, we use the unit Tg N /yr/yr (Tg N yr$^{-2}$).

**Comment 32**: L.359. These high values cannot be seen in Figure 7, which has an upper boundary of >10 g N m2 yr-1 (resulting in an even colouring of all China since the 2000s).I suggest changing the colour scale to be able to see these high values.

Response: The figure has been revised according to your suggestion.

**Comment 33**: Figure 8. I wonder if this figure is necessary, given the high uncertainty in the data, and the fact that homogeneous N application rates are used for each country…

Response: We have moved Figure 8 to the supplementary Figure S1.

**Comment 34**: Figure 13. Similar graphs showing the data per unit area would be welcome

Response: Figure 13 has been revised to further show the N input rate-GDP per capita relationships. The following statements were also added:

"The N input rate-GDP per capita relationships also generally fell into the three groups (Fig. 13). But a notable phenomenon is that N input rate in Korea and Japan was much higher than other regions in almost all the three agricultural sectors (Fig. 13). This is also reported by (Lim et al., 2021) and they attributed it to decrease of arable land area, high fertilizer input and especially large manure inputs, although fertilizer input in Korea had been considerably reduced." L463-468.

**References:**

Conant, R. T., Berdanier, A. B. & Grace, P. R. Patterns and trends in nitrogen use and nitrogen recovery efficiency in world agriculture. Global Biogeochemical Cycles 27, 558-566, doi: 10.1002/Gbc.20053 (2013).

Einarsson, R., Sanz-Cobena, A., Aguilera, E., Billen, G., Garnier, J., van Grinsven, H.J.M., Lassaletta, L., 2021. Crop production and nitrogen use in European cropland and grassland 1961–2019. Scientific Data 8, 288.

Herridge, D. F., Peoples, M. B., and Boddey, R. M.: Global inputs of biological nitrogen fixation in agricultural systems, Plant Soil, 311, 1–18, https://doi.org/10.1007/s11104-008-9668-3, 2008.

Holland, E. A., Lee-Taylor, J., Nevison, C., and Sulzman, J. M.: Global N Cycle: fluxes and N2O mixing ratios originating from human activity, ORNL DAAC, 2005.

Lassaletta, L., Billen, G., Grizzetti, B., Anglade, J. & Garnier, J. 50 year trends in nitrogen use efficiency of world cropping systems: the relationship between yield and nitrogen input to cropland. Environmental Research Letters 9, 105011 (2014).

Lim, J. Y., Islam Bhuiyan, M. S., Lee, S. B., Lee, J. G., and Kim, P. J.: Agricultural nitrogen and phosphorus balances of Korea and Japan: Highest nutrient surplus among OECD member countries, Environmental Pollution, 286, 117353, https://doi.org/10.1016/j.envpol.2021.117353, 2021.

Monfreda, C., Ramankutty, N., Foley, J.A., 2008. Farming the planet: 2. Geographic distribution of crop areas, yields, physiological types, and net primary production in the year 2000. Global Biogeochemical Cycles 22.

Xu, R., Tian, H., Pan, S., Dangal, S.R.S., Chen, J., Chang, J., Lu, Y., Skiba, U.M., Tubiello, F.N., Zhang, B., 2019. Increased nitrogen enrichment and shifted patterns in the world's grassland: 1860–2016. Earth Syst. Sci. Data 11, 175-187.

Zhang, B., Tian, H., Lu, C., Dangal, S.R.S., Yang, J., Pan, S., 2017. Global manure nitrogen production and application in cropland during 1860–2014: a 5 arcmin gridded global dataset for Earth system modeling. Earth Syst. Sci. Data 9, 667-678.

Zhang, L., Tian, H., Shi, H., Pan, S., Chang, J., Dangal, S., Qin, X., Wang, S., Tubiello, F., Canadell, J., Jackson, R., 2022. A 130-year global inventory of methane emissions from livestock: trends, patterns, and drivers. Global Change Biol.

Zhang, X., Davidson, E.A., Mauzerall, D.L., Searchinger, T.D., Dumas, P., Shen, Y., 2015. Managing nitrogen for sustainable development. Nature 528, 51-59.